# On-Surface Synthesis of Polypyridine: Strain Enforces Extended Linear Chains

Laerte L. Patera [1,2,*], Josef Amler [1] and Jascha Repp [1]

1   Institute of Experimental and Applied Physics, University of Regensburg, 93053 Regensburg, Germany; j-amler@gmx.de (J.A.); jascha.repp@ur.de (J.R.)
2   Catalysis Research Center, Department of Chemistry, Technical University of Munich, 85748 Garching, Germany
*   Correspondence: laerte.patera@tum.de

**Abstract:** Strain-induced on-surface transformations provide an appealing route to steer the selectivity towards desired products. Here, we demonstrate the selective on-surface synthesis of extended all-*trans* poly(2,6-pyridine) chains on Au(111). By combining high-resolution scanning tunneling and atomic force microscopy, we revealed the detailed chemical structure of the reaction products. Density functional theory calculations indicate that the synthesis of extended covalent structures is energetically favored over the formation of macrocycles, due to the minimization of internal strain. Our results consolidate the exploitation of internal strain relief as a driving force to promote selective on-surface reactions.

**Keywords:** on-surface synthesis; AFM; STM; DFT calculations

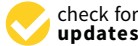



## 1. Introduction

On-surface synthesis of carbon-based nanostructures has rapidly emerged as a fascinating method for the synthesis of nanomaterials, with structures and functionalities not achievable by wet chemistry [1–4]. One of the most used reaction schemes is based on the Ullmann coupling, where the dehalogenation of suitable precursors allows creating reactive sites within the molecules, which initiate the polymerization reaction [5]. Control over the side reactions can be further achieved through molecule–substrate interactions [6–10]. For instance, substrate symmetry has been exploited to tune the major reaction pathway for 4,4″-dibromo-meta-terphenyl precursors. The close-packed Cu(111) surface was observed to steer the formation of covalent macrocycles, while Cu(110) favors the growth of extended structures [11,12]. Substrate-induced internal strain can drive the skeletal rearrangement of an extended 1D metal–organic chain, enabling the formation of an energetically favorable registry with the Cu(111) substrate [13]. Intramolecular strain relief represents another appealing approach to promote the formation of desired molecular products as the on-surface planarization of distorted polycyclic aromatic hydrocarbons [14] and the synthesis of nanographene [15,16]. Aiming at achieving extended nanostructures, intermolecular steric effects have been shown to play a crucial role in driving the sequential cyclohydrogenation reaction of polyantracene oligomers for the synthesis of graphene nanoribbons [17]. Here, we demonstrate the selective synthesis of poly(2,6-pyridine) structures on the Au(111) surface by using 6,6″-dibromo-2,2′:6′,2″-terpyridine (DBTP) molecules as precursors. By combining high-resolution scanning tunneling and atomic force microscopy (STM/AFM) with density functional theory (DFT) calculations, we show that internal strain relief favors the formation of extended covalent chains composed of all-*trans* pyridines, while all-*cis* pyridine macrocycles represent only a minority of the surface products.

## 2. Materials and Methods

Experiments were carried out with a low-temperature scanning tunneling and atomic force microscope (CreaTec Fischer & Co. GmbH, Berlin, Germany) equipped with a qPlus tuning fork [18] (quality factor $Q \approx 5 \times 10^4$, stiffness $k \approx 1.8$ kN/m, resonance frequency $f_{res} \approx 29$ kHz) operating in the frequency modulation mode at a base temperature of 9.1 K under ultrahigh vacuum conditions ($p \approx 2 \times 10^{-10}$ mbar). Bias voltages ($V$) refer to the sample with respect to the tip. The Au(111) surface was prepared by sputtering (Ne$^+$, 1 keV) and annealing cycles (770 K). DBTP molecules were sublimed onto the Au(111) sample surface kept at 300 K, followed by annealing to 470 K to initiate polymerization. Finally, the sample was transferred to the microscope and cooled down to 9.1 K for experiments. AFM images were recorded in constant-height mode, where positive z-offset ($\Delta z$) values indicate a retraction of the tip after opening the feedback loop with respect to the STM setpoint above the clean Au(111) surface. The oscillation amplitude $A$ was kept constant at $A = 0.5$ Å. CO molecules were dosed on the cold sample inside the microscope ($T < 12$ K) and used for tip functionalization [19].

Calculations were performed using the ORCA (4.0.1) program package (Max-Planck-Institut für Kohlenforschung, Mülheim an der Ruhr, Germany) to optimize geometries [20]. The PBE0 density functional, in combination with the correlation-consistent double-zeta (cc-pVDZ) basis set, was used. To speed up calculations, the RIJCOSX approximation [21] (in combination with the def2/J auxiliary basis set) was used. To quantitatively compare the energetics of the all-*trans* and all-*cis* configurations, for the latter, an H$_2$ molecule has been added to the computation.

## 3. Results and Discussion

Given the structure of the DBTP precursors, possible products of the Ullmann coupling are extended covalent chains and covalent macrocycles (Scheme 1). Linear chains are achieved through all-*trans* orientation of adjacent pyridine groups, while macrocycles can be obtained by the all-*cis* conformation.

**Scheme 1.** Chemical structure of the 6,6''-dibromo-2,2':6',2''-terpyridine (DBTP) precursor and two possible products of the on-surface synthesis: all-*trans* chain (left) and all-*cis* macrocycle (right).

Surface annealing to 470 K induces the full debromination of the DBTP precursors, initiating the Ullmann coupling. Figure 1 shows typical STM images acquired after annealing a submonolayer (Figure 1a,b) and a multilayer (Figure 1c) of DBTP. In both cases, the main reaction products consist of extended polymeric chains (up to 100 nm long). Upon polymerization of a submonolayer of DBTP precursors, kink sites were observed (Figure 1a), resulting in chain bending by 60 and 120°. Such sites originate from the presence of *cis* pyridine units within all-*trans* polymeric chains.

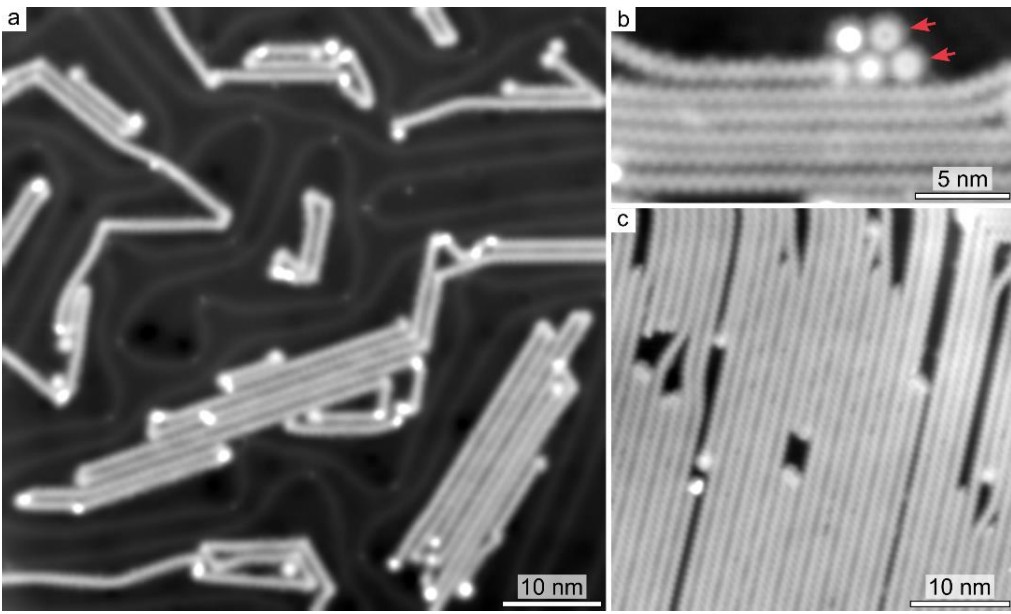

**Figure 1.** Constant-current STM images after annealing to 470 K of a Au(111) surface covered by a submonolayer (**a**,**b**) and multilayer (**c**) of DBTP. (**b**) Arrows highlight macrocycles. Measurement parameters: (**a**) tunneling current $I = 1.9$ pA, sample bias voltage $V = 1.0$ V; (**b**) $I = 5$ pA, $V = 1.0$ V; (**c**) $I = 2.6$ pA, $V = 1.0$ V.

Remarkably, despite polymers composed of a random mixture of *trans* and *cis* conformations being expected from geometrical considerations, the large majority of reaction products consists of all-*trans* structures. Furthermore, the density of kink sites is even more suppressed at larger polymer coverages (Figure 1c), pointing out also the role of interchain steric effects in guiding the growth of aligned polymers [22]. While most of the reaction products consist of extended chains upon annealing to 450–500 K, cyclic structures are also found on the surface (Figure 1b). The lateral size of about 13 Å matches the size of the macrocycles composed of six all-*cis* pyridine units, indicated in Scheme 1.

Figure 2a shows a submolecularly resolved STM image of the extended products. The zig-zag appearance of the linear segments suggests the formation of regular polypyridine chains (Scheme 1). Detailed insight into the chemical structure is provided by AFM imaging with CO-functionalized tips [19]. The contrast in the AFM frequency shift ($\Delta f$) image (Figure 2b) clarifies the presence of covalent bonds between DBTP molecules, resulting in the formation of extended all-*trans* poly(2,6-pyridine) structures. While poly-phenylene chains typically exhibit a large torsional angle between adjacent monomer units, due to steric repulsion between the H atoms [23,24], H bonding favors a structural planarization for all-*trans* poly(2,6-pyridine), as indicated by the rather uniform $\Delta f$ contrast along the pyridine units. Coupling between individual chains is suggested to be mediated by Br adatoms [25,26], which favor either local in-phase or anti-phase alignments (Figure 2a). AFM imaging of a kink site (Figure 2c) reveals that the presence of two adjacent pyridine groups in *cis* conformation gives rise to a 60° change in the chain orientation. Notably, pyridines at the kink site exhibit a different $\Delta f$ contrast, indicating an out-of-plane tilt [24,27–31], with N atoms lying closer to the substrate. The origin of this structural

distortion can be attributed to steric hindrance between pairs of adjacent pyridinic N atoms, pairs of H atoms facing each other from adjacent pyridine rings, and the lack of stabilizing C-H-N′ interactions [32–34]. This observation indicates that the presence of pyridine groups in *cis* conformation within the polymer induces substantial internal strain.

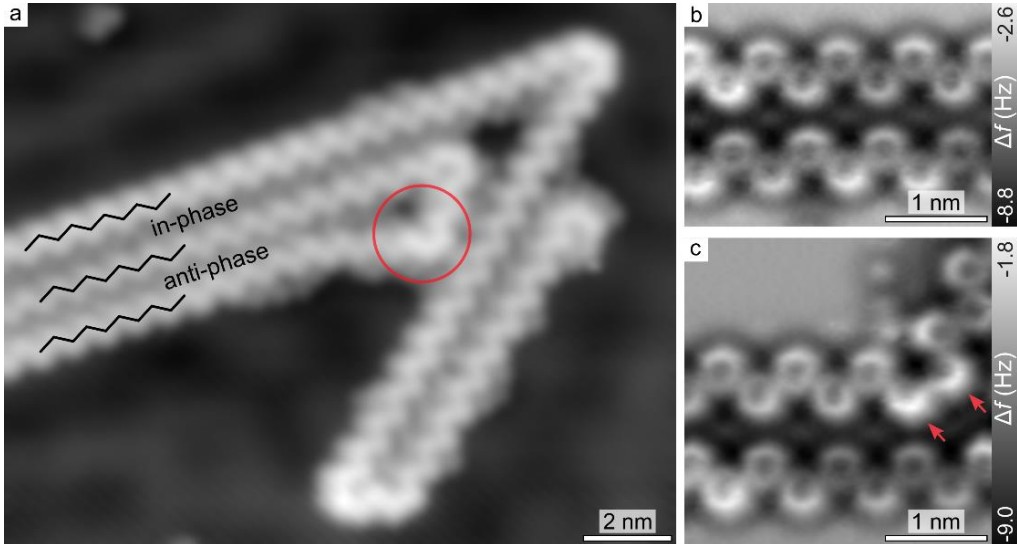

**Figure 2.** (**a**) Constant-current STM image after annealing to 470 K. Colored circle indicates a kink site. Black lines highlight the local in-phase and anti-phase chain alignment (*I* = 2.1 pA, *V* = 1.0 V). (**b**,**c**) Constant-height Δ*f* AFM images acquired with a CO-functionalized tip at Δ*z* = −1.5 Å, given with respect to an STM setpoint of (*I* = 1.5 pA, *V* = 0.1 V). (**b**) Anti-phase chain alignment. (**c**) Kink site. Arrows indicate the protruding sides of tilted pyridine groups.

Furthermore, due to the low energy barrier for pyridine rotation around the C-C bond in the gas phase, three isomers are expected to be adsorbed on the Au(111) surface. Nevertheless, the experimental observation of all-*trans* chains indicates that the barrier for pyridine rotation can be easily overcome also on the Au(111) surface (especially during the annealing step to 450–500 K), resulting in the transformation of most of the precursor molecules into the all-*trans* configuration.

In order to further support this hypothesis, we performed density functional theory (DFT) calculations for oligomers composed of six pyridine groups in all-*trans* (Figure 3a,b) and all-*cis* (Figure 3c,d) conformations. Gas-phase calculations were employed to elucidate the role of internal strain in guiding the formation of all-*trans* polymers. The all-*trans* configuration exhibits a rather planar geometry, while all-*cis* is characterized by a strong structural distortion, lying ≈ 2.2 eV higher in energy with respect to the all-*trans* oligomer. Furthermore, we performed calculations for an oligomer having two *cis* pyridine groups in the middle (Figure 3e,f). Such relaxed structure is energetically less favorable (≈0.25 eV) than the all-*trans* configuration. While the adsorption on the gold surface is expected to mitigate such distortion [24], the calculations indicate how the *trans* configuration is energetically favored, representing the main driving force behind the selective on-surface formation of extended polypyridine chains.

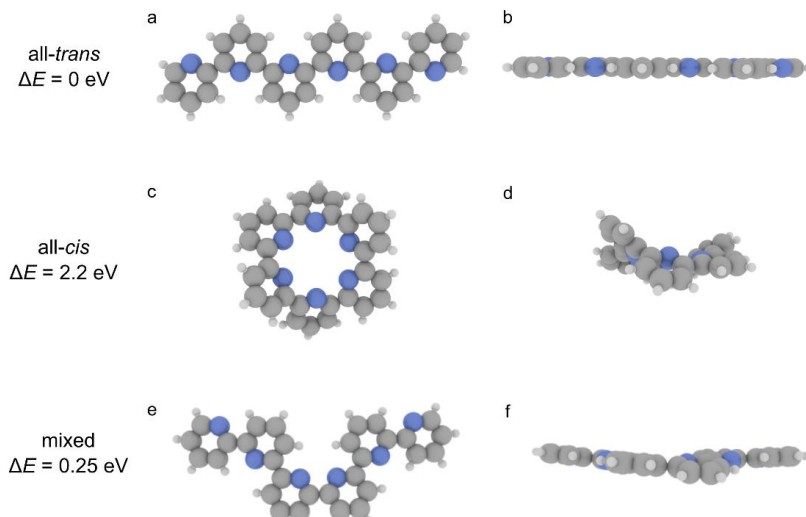

**Figure 3.** Gas-phase optimized geometries of pyridine oligomers in all-*trans* (**a**,**b**), all-*cis* (**c**,**d**), and mixed (**e**,**f**) configurations, respectively. (**a**,**c**,**e**) Top and (**b**,**d**,**f**) side views. Grey, blue, and white spheres represent carbon, nitrogen, and hydrogen atoms, respectively.

## 4. Conclusions

We reported the synthesis of extended polypyridine chains on Au(111) through Ullmann coupling. By combining high-resolution STM and AFM, we revealed the chemical structure of the products. While both extended all-*trans* polypyridine chains and macrocycles are observed, the latter represent only a minority of the products. This observation is rationalized by the structural distortion induced by the presence of *cis* pyridines within covalent structures. Our results demonstrate how internal strain can be successfully exploited to steer the selectivity in an on-surface reaction, consolidating the use of strain-driven reactions to promote the formation of desired molecular products.

**Author Contributions:** Conceptualization and supervision, L.L.P. and J.R.; formal analysis, L.L.P.; performing experiment, L.L.P. and J.A.; writing—original draft preparation, L.L.P.; writing—review and editing, L.L.P. and J.R.; visualization, L.L.P. All authors have read and agreed to the published version of the manuscript.

**Funding:** This research was funded by the Deutsche Forschungsgemeinschaft (DFG, German Research Foundation), grant numbers: IDs RE2669/6 and PA3628/1.

**Data Availability Statement:** Data are contained within the article.

**Conflicts of Interest:** The authors declare no conflict of interest.

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
