# Peer review of "On-Surface Synthesis of Polypyridine: Strain Enforces Extended Linear Chains"

_chemistry, doi:10.3390/chemistry4010009_

Round 1

Reviewer 1 Report

The manuscript by Patera and coworkers discusses the on-surface synthesis of polypyridine chains from dihalogenated  terpyridine monomers. To create these polymers the Ullman coupling reaction on Au(111) surface under ultra-high vacuum conditions was used and its products were imaged with the STM technique. The experimental investigations described in the manuscript were accompanied by the corresponding quantum density functional theory (DFT) calculations. The Authors demonstrated that the main factor which pushes the molecules to create chains instead of ring oligomers is the larger internal strain occurring in structures of the latter type. The work is interesting, as it provides some useful information on how the self-assembly and polymerization on surfaces can be directed by a suitable choice of the building block. The manuscript is clearly written and its conclusions are supported  by the reported results. In summary, I recommend publication of the work provided the following minor points have been addressed:

1)  Some more details about the DFT calculations should be provided. What assumptions were made and what software was used?

2) Can the Authors comment om the effect of temperature on the observed structure formation. Was there any influence of temperature on the number of cyclic oligomers. If I am correct, there four of them in Fig. 1b (top-right).  

3) It would be useful to compare the energy of a single all-trans chain (even for two trans tectons) with the energy of a random chain ( cis-trans connection). This might bring the answer to why only the trans chains were observed in the STM.

4) Was there observed the rotation of the terminal pyridine groups around the C-C bond linking this group with the central one? If so, the chirality question arises – if one group flips over then two surface enantiomers can be created.

Author Response

Reviewer 1: The manuscript by Patera and coworkers discusses the on-surface synthesis of polypyridine chains from dihalogenated terpyridine monomers. To create these polymers the Ullman coupling reaction on Au(111) surface under ultra-high vacuum conditions was used and its products were imaged with the STM technique. The experimental investigations described in the manuscript were accompanied by the corresponding quantum density functional theory (DFT) calculations. The Authors demonstrated that the main factor which pushes the molecules to create chains instead of ring oligomers is the larger internal strain occurring in structures of the latter type. The work is interesting, as it provides some useful information on how the self-assembly and polymerization on surfaces can be directed by a suitable choice of the building block. The manuscript is clearly written and its conclusions are supported by the reported results. In summary, I recommend publication of the work provided the following minor points have been addressed:

Author reply: We thank the reviewer for his/her careful reading of our manuscript, the overall positive remarks, and further comments/suggestions.

Reviewer 1: 1) Some more details about the DFT calculations should be provided. What assumptions were made and what software was used?

Author reply: The details of the calculations are included in the “Materials and Methods” section:
“Calculations were performed using the ORCA (4.0.1) program package to optimize geometries [20]. The PBE0 density functional, in combination with the correlation-consistent double-zeta (cc-pVDZ) basis set, was used. To speed up the calculations, the RIJCOSX approximation [21] (in combination with the def2/J auxiliary basis set) was used. To quantitatively compare the energetics of the all-trans and all-cis configurations, for the latter an H2 molecule has been added to the computation.”

Reviewer 1: 2) Can the Authors comment om the effect of temperature on the observed structure formation. Was there any influence of temperature on the number of cyclic oligomers. If I am correct, there four of them in Fig. 1b (top-right).  

Author reply: In the temperature range we explored (450-500 K), no relevant difference has been seen in the occurrence of cyclic oligomers. As reported in the manuscript, the main parameter which influences the yield of cyclic oligomers is the coverage of precursors, since at large coverages steric effects further steer the formation of all-trans species. The following sentence has been added to the manuscript: “While upon annealing to 450-500 K most of the reaction products consist of extended chains, cyclic structures are also found on the surface.”

Reviewer 1: 3) It would be useful to compare the energy of a single all-trans chain (even for two trans tectons) with the energy of a random chain (cis-trans connection). This might bring the answer to why only the trans chains were observed in the STM.

Author reply: We agree with the referee that a comparison of all-trans and all-cis species with a mixed one could be useful to better understand the energetic driving force behind the formation of all-trans chains. For this reason, we performed further calculations, especially of an oligomer having two cis pyridine groups in the middle (added now in Figure 3). Such relaxed structure lies about 0.25 eV higher in energy than the all-trans configuration, confirming that the trans configuration is energetically favored over local cis one. The following sentence has been added to the manuscript: “Furthermore, we performed calculations for an oligomer having two cis pyridine groups in the middle (Figure 3c). Such relaxed structure is energetically less favorable (≈ 0.25 eV) than the all-trans configuration.”

4) Was there observed the rotation of the terminal pyridine groups around the C-C bond linking this group with the central one? If so, the chirality question arises – if one group flips over then two surface enantiomers can be created.

Author reply: We thank the referee for helping us to clarify this point. In principle, due to the low energy barrier for pyridine rotation around the C-C bond in the gas phase, a total of three isomers is expected to adsorb on the surface. In this context, the experimental observation of all-trans chains strongly indicates that the barrier for pyridine rotation can be easily overcome also on the Au(111) surface (especially during the annealing step to 470 K), leading to the transformation of most of the precursors molecules in to the all-trans configuration. The following sentence has been added to the manuscript: “Furthermore, due to the low energy barrier for pyridine rotation around the C-C bond in the gas phase, three isomers are expected to adsorb on the Au(111) surface. Nevertheless, the experimental observation of all-trans chains indicates that the barrier for pyridine rotation can be easily overcome also on the Au(111) surface (especially during the annealing step to 450-500 K), leading to the transformation of most of the precursors molecules in to the all-trans configuration.”

Reviewer 2 Report

I find the idea of the manuscript interesting and potentially suitable for this journal. However, the associated DFT calculations that should help explain the experiments have to be improved along these lines:

1) It does not make much sense to compare energies of 2 structures with a different number of atoms (namely, 2 extra hydrogens). As the structure with more atoms will certainly have lower energy. Calculations have to be repeated with an equal number of atoms. Why not make periodic boundary conditions for trans configuration?

2) Some more details are missing in the methodology to enable reproducibility.

3) Ideally, calculations should be performed with molecules on the surface. Such calculations are routinely done nowadays and I don't see any reason why they were not done here too.

Author Response

Reviewer 2: I find the idea of the manuscript interesting and potentially suitable for this journal. However, the associated DFT calculations that should help explain the experiments have to be improved along these lines:

Author reply: We thank the referee for the positive assessment of our study.

Reviewer 2: 1) It does not make much sense to compare energies of 2 structures with a different number of atoms (namely, 2 extra hydrogens). As the structure with more atoms will certainly have lower energy. Calculations have to be repeated with an equal number of atoms. Why not make periodic boundary conditions for trans configuration?

Author reply: We fully agree with the referee that comparison is reasonable only when structures have the very same number of atoms. In our case, to quantitatively compare the energetics of the all-trans and all-cis configurations, for the latter an H2 molecule has been added to the computation. In this way, we can compare structures with exactly the same atom number and obtain that the all-trans configuration lies about 2.2 lower in energy that the all-cis plus H2.
While implementing periodic boundaries would be definitely interesting to explore the energetics of extended chains, our approach allows for direct energetic comparison between all-trans and all-cis structures.

Reviewer 2: 2) Some more details are missing in the methodology to enable reproducibility.

Author reply: We have now added further details about the calculation: “To speed up the calculations, the RIJCOSX approximation [21] (in combination with the def2/J auxiliary basis set) was used.”

Reviewer 2: 3) Ideally, calculations should be performed with molecules on the surface. Such calculations are routinely done nowadays and I don't see any reason why they were not done here too.

Author reply: We agree with the referee that calculations including the surface would provide a better comparison with the experiments, and that the use of simplified gas-phase calculations should be properly justified. We would like to stress that, for our system, gas-phase calculations can indeed provide useful information to support the understanding of experimental data. In fact, gas-phase calculations clearly allow elucidating the role of internal strain in guiding the formation of all-trans polymers. Notably, this ultimately shows that the energetic driving force for the formation of extended covalent structures is already given by minimization of internal strain. This is an important difference to most studies, where substrate-induced strain was exploited to obtain a specific reaction pathway. For these reasons, while extensive calculations including the surface would be definitely helpful to elucidate the role of the substrate in steering the adsorption configuration of polypyridine, we believe that gas phase calculations, despite their simplicity, are well suited to provide useful insights in relation to the scope of our study. The following sentence has been added to the manuscript: “Gas-phase calculation were employed to elucidate the role of internal strain in guiding the formation of all-trans polymers.”

Round 2

Reviewer 2 Report

I still think it would be much better if the authors implemented suggested changes but the manuscript seems technically correct and it can be accepted in present form.